# ProCLIP: Product Space Multimodal Contrastive Alignment

**Jiakai Chen**
Siebel School of Computing and Data Science
University of Illinois Urbana-Champaign
Urbana, IL 61801, USA
jiakaic3@illinois.edu

**Hangke Sui**
Electrical and Computer Engineering
University of Illinois Urbana-Champaign
Urbana, IL 61801, USA
hangkes2@illinois.edu

## ABSTRACT

Contrastive learning has become a dominant paradigm for multimodal representation learning, aligning paired observations (e.g., image–text) in a shared embedding space. Most existing approaches, however, adopt a single latent geometry, implicitly assuming that one manifold can capture the heterogeneous structure of multimodal semantics. In this work, we propose a mixed-curvature product embedding space that combines hyperbolic, Euclidean, and spherical factors within a unified latent manifold. We equip this product space with a weighted product metric and define a geometry-aware CLIP objective by measuring cross-modal similarity, yielding a drop-in replacement for standard cosine-based alignment while respecting manifold constraints. We instantiate the ProCLIP framework with lightweight manifold-specific projection heads and evaluate it on standard image-text retrieval benchmarks. Empirically, mixed-curvature product embeddings consistently improve cross-modal retrieval and alignment over single-manifold baselines, and our analyses highlight regimes in which heterogeneous curvature provides the largest gains.

## 1 INTRODUCTION

Learning shared representations across modalities, such as vision–language, video–audio, and video–motion, is a central problem in modern multimodal representation learning. A dominant paradigm is *contrastive learning*, exemplified by CLIP-style objectives that align paired samples while repelling mismatched pairs in a shared embedding space (Radford et al., 2021; Jia et al., 2021; Li et al., 2022). Despite remarkable empirical success, most contrastive multimodal systems embed all modalities into a *single* latent geometry, which is typically Euclidean with cosine similarity on the unit sphere (Oord et al., 2018), implicitly assuming that a single manifold can simultaneously accommodate heterogeneous semantic structures arising across modalities and levels of abstraction.

This "single-manifold" assumption is often misaligned with the geometry of real-world semantics. HyCoCLIP (Pal et al., 2024) argues that vision–language semantics contain both inter-modal and intra-modal hierarchical structure. Multimodal concepts frequently exhibit *mixed* structural properties: hierarchical relations and entailment (e.g., animal $\rightarrow$ dog), cyclic or angular variation (e.g., phase), and locally linear continuous factors (e.g., intensity), as illustrated in Figure 1. A single constant-curvature geometry may therefore over-constrain the representation space, forcing distinct semantic factors to compete for the same geometric degrees of freedom. For example, consider an image of a yellow puppy and an image of a yellow kitten. They are *close* in terms of color and overall visual tone, yet *far* in terms of category identity (dog vs. cat). At the same time, both concepts are semantically nested under a shared superclass animal, exhibiting an explicit hierarchical relation between the abstract concept (animal) and the specific instances (dog, cat). This illustrates that real-world semantics is often difficult to capture with a single uniform geometry.

In the single-modality setting, prior work (Gu et al., 2018) has suggested that allowing *mixed curvature* can better capture heterogeneous structure than any single geometry alone (e.g., combining hyperbolic and spherical components for different relational patterns).

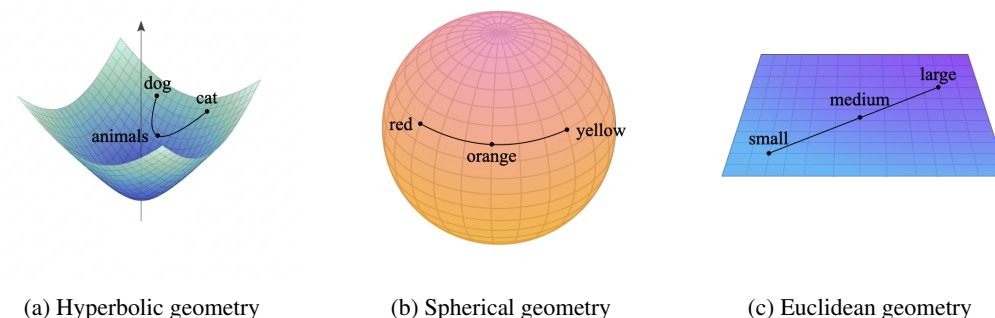

(a) Hyperbolic geometry      (b) Spherical geometry      (c) Euclidean geometry

Figure 1: **Illustration of heterogeneous semantic structures and their natural geometries**. Hyperbolic space captures hierarchical relations (e.g., *animal → dog/cat*), spherical space captures directional or angular similarity (e.g., color hue such as red → orange → yellow), and Euclidean space captures continuous linear variation (e.g., size or intensity). These examples motivate modeling multimodal representations in a mixed-curvature product space rather than a single uniform geometry.

Motivated by this perspective, we propose ProCLIP, leveraging *mixed-curvature product space* for multimodal contrastive learning. Concretely, we embed each modality into a shared product manifold $\mathcal{Z} = \mathbb{H}_c^{d_H} \times \mathbb{R}^{d_E} \times \mathbb{S}_{R_S}^{d_S}$, equipped with a weighted product metric that aggregates distances across components. Heuristically, this construction decomposes the latent geometry into components with complementary inductive biases: the **hyperbolic factor (negative curvature)** is suited for capturing hierarchical relations, the **spherical factor (positive curvature)** emphasizes directional similarity under normalization, and the **Euclidean factor (zero curvature)** provides a flexible substrate for continuous variations.

Our main contributions are: (a) we propose a mixed-curvature product embedding space for multimodal alignment by combining learnable-curvature factors (hyperbolic, Euclidean, and spherical) into a single shared latent manifold; (b) we develop a geometry-aware, CLIP-style contrastive objective using a weighted product metric, serving as a simple drop-in replacement for cosine similarity while respecting manifold constraints; and (c) we implement the resulting framework with lightweight manifold-specific projection heads and validate it on standard multimodal retrieval benchmarks.

## 2    GEOMETRY-AWARE MULTIMODAL ALIGNMENT VIA MIXED-CURVATURE PRODUCT SPACES

We consider $N$ paired samples $\{(\mathbf{x}_i, \mathbf{y}_i)\}_{i=1}^N$, where $\mathbf{x}_i \in \mathbb{R}^{d_1}$ and $\mathbf{y}_i \in \mathbb{R}^{d_2}$ are semantically aligned observations from two modalities. The objective of contrastive learning is to learn encoders

$$\mathbf{f}_{\theta_1} : \mathbb{R}^{d_1} \to \mathbb{R}^r, \qquad \mathbf{f}_{\theta_2} : \mathbb{R}^{d_2} \to \mathbb{R}^r,$$

with modality-specific parameters $\theta_1, \theta_2$, such that paired inputs are mapped to similar representations in a shared $r$-dimensional embedding space while non-paired inputs remain dissimilar.

In this work, we generalize the shared embedding space from a single Euclidean space to a *mixed-curvature product manifold* $\mathcal{Z}$. Accordingly, we view each encoder as mapping into $\mathcal{Z}$:

$$\mathbf{z}_i = \mathbf{f}_{\theta_1}(\mathbf{x}_i) \in \mathcal{Z}, \qquad \tilde{\mathbf{z}}_i = \mathbf{f}_{\theta_2}(\mathbf{y}_i) \in \mathcal{Z}. \tag{1}$$

We use product manifold similarity and distances on $\mathcal{Z}$ and optimize a contrastive objective to align $(\mathbf{z}_i, \tilde{\mathbf{z}}_i)$ against negatives $(\mathbf{z}_i, \tilde{\mathbf{z}}_j)$ for $j \neq i$.

### 2.1    PRODUCT SPACE OF MIXED CURVATURE

**Hyperbolic component (Lorentz model (Nickel & Kiela, 2018)).** Let $\mathbb{R}^{r_H+1}$ be equipped with the Minkowski (Lorentz) metric $\langle \mathbf{u}, \mathbf{v} \rangle_L = \sum_{k=1}^{r_H} u_k v_k - u_0 v_0 = \mathbf{u}_{1:r_H}^\top \mathbf{v}_{1:r_H} - u_0 v_0$. For curvature

parameter $c > 0$, define the $r_H$-dimensional hyperbolic space as

$$\mathbb{H}_c^{r_H} = \left\{ \mathbf{z} \in \mathbb{R}^{r_H+1} : \langle \mathbf{z}, \mathbf{z} \rangle_L = -\frac{1}{c}, \; z_0 > 0 \right\}. \tag{2}$$

For $\mathbf{z}_1, \mathbf{z}_2 \in \mathbb{H}_c^{r_H}$, the geodesic distance is $d_{\mathbb{H}}(\mathbf{z}_1, \mathbf{z}_2) = \frac{1}{\sqrt{c}} \operatorname{arcosh}(-c \langle \mathbf{z}_1, \mathbf{z}_2 \rangle_L)$. Hyperbolic geometry provides an inductive bias for hierarchical or tree-like structure.

**Spherical component.** Let $\mathbb{S}^{r_S} = \left\{ \mathbf{s} \in \mathbb{R}^{r_S+1} : \|\mathbf{s}\|_2 = R \right\}$ be the sphere with geodesic distance $d_{\mathbb{S}}(\mathbf{s}_1, \mathbf{s}_2) = R \arccos\left(\frac{\langle \mathbf{s}_1, \mathbf{s}_2 \rangle}{R^2}\right), \mathbf{s}_1, \mathbf{s}_2 \in \mathbb{S}^{r_S}$. Spherical geometry captures directional similarity under normalization, which is closely related to cosine similarity.

**Euclidean component.** We include a Euclidean factor $\mathbb{R}^{r_E}$ with standard distance $d_{\mathbb{E}}(\mathbf{e}_1, \mathbf{e}_2) = \|\mathbf{e}_1 - \mathbf{e}_2\|_2, \mathbf{e}_1, \mathbf{e}_2 \in \mathbb{R}^{r_E}$, to capture locally linear and additive variations.

**Mixed-curvature product space.** We define the shared latent space as the Cartesian product

$$\mathcal{Z} := \mathbb{H}_c^{r_H} \times \mathbb{R}^{r_E} \times \mathbb{S}^{r_S}, \qquad r = r_H + r_E + r_S, \tag{3}$$

where $r$ denotes the total embedding dimension. Accordingly, each embedding decomposes as

$$\mathbf{z} = (\mathbf{z}^{(H)}, \mathbf{z}^{(E)}, \mathbf{z}^{(S)}), \quad \mathbf{z}^{(H)} \in \mathbb{H}_c^{r_H}, \; \mathbf{z}^{(E)} \in \mathbb{R}^{r_E}, \; \mathbf{z}^{(S)} \in \mathbb{S}^{r_S}.$$

We equip $\mathcal{Z}$ with a weighted product metric. For $\mathbf{z}_1, \mathbf{z}_2 \in \mathcal{Z}$, define

$$d_{\mathcal{Z}}^2(\mathbf{z}_1, \mathbf{z}_2) := \alpha_H \, d_{\mathbb{H}}^2\big(\mathbf{z}_1^{(H)}, \mathbf{z}_2^{(H)}\big) + \alpha_E \, \|\mathbf{z}_1^{(E)} - \mathbf{z}_2^{(E)}\|_2^2 + \alpha_S \, d_{\mathbb{S}}^2\big(\mathbf{z}_1^{(S)}, \mathbf{z}_2^{(S)}\big), \tag{4}$$

where $\alpha_H, \alpha_E, \alpha_S > 0$ control the relative contribution of each factor.

**Remark 1 (Learnable weights as effective curvature parameters)** *On constant-curvature manifolds, scaling the metric by a positive factor is equivalent to rescaling the curvature. In particular, hyperbolic distances satisfy $d_{\mathbb{H}_c}(\mathbf{z}_1, \mathbf{z}_2) = \frac{1}{\sqrt{c}} d_{\mathbb{H}_1}(\mathbf{z}_1, \mathbf{z}_2)$, so learning $\alpha_H$ in equation 4 is equivalent (up to reparameterization) to learning an* effective *curvature scale for the hyperbolic factor. Analogous interpretations apply to other constant-curvature factors.*

## 2.2   Product-space contrastive alignment

We define similarity by the negative squared product distance $s(\mathbf{z}, \tilde{\mathbf{z}}) := -d_{\mathcal{Z}}^2(\mathbf{z}, \tilde{\mathbf{z}})$. We then adopt a CLIP-style objective

$$\mathcal{L} = -\sum_{i=1}^{N} \log \frac{\exp(s(\mathbf{z}_i, \tilde{\mathbf{z}}_i)/\tau)}{\sum_{j=1}^{N} \exp(s(\mathbf{z}_i, \tilde{\mathbf{z}}_j)/\tau)} - \sum_{i=1}^{N} \log \frac{\exp(s(\mathbf{z}_i, \tilde{\mathbf{z}}_i)/\tau)}{\sum_{j=1}^{N} \exp(s(\mathbf{z}_j, \tilde{\mathbf{z}}_i)/\tau)}. \tag{5}$$

**Implementation (factorized heads).** We parameterize each encoder as a backbone followed by factorized projection heads $\mathbf{f}_{\theta_m}(\cdot) = \big(\mathbf{f}_{\theta_m}^{(H)}(\cdot), \; \mathbf{f}_{\theta_m}^{(E)}(\cdot), \; \mathbf{f}_{\theta_m}^{(S)}(\cdot)\big), m \in \{1, 2\}$. The Euclidean head outputs $\mathbf{z}^{(E)} \in \mathbb{R}^{r_E}$ directly; the spherical head outputs a vector in $\mathbb{R}^{r_S+1}$ and normalizes it onto $\mathbb{S}^{r_S}$. For the hyperbolic head, a standard construction is to predict a tangent vector $\mathbf{u} \in T_{\mathbf{o}}\mathbb{H}_c^{r_H}$ at a fixed origin $\mathbf{o}$ and map it to the manifold via the exponential map $\mathbf{z}^{(H)} = \exp_{\mathbf{o}}(\mathbf{u}) \in \mathbb{H}_c^{r_H}$, ensuring manifold constraints are respected.

## 2.3   Product space vs. single-manifold embeddings

Single-manifold embeddings impose a uniform geometric inductive bias across all latent degrees of freedom (e.g., purely Euclidean, purely spherical, or purely hyperbolic). This can be restrictive when the semantics exhibit heterogeneous structure (hierarchical, angular, and locally linear components simultaneously). In contrast, the product construction equation 3 induces a block-structured geometry in which distinct semantic factors can be represented in geometrically appropriate components. The weighted product metric equation 4 further allows the model to adapt the relative strength of these inductive biases (Remark 1), providing a flexible yet principled geometric backbone for contrastive alignment.

## 3 EXPERIMENTS

We evaluate the proposed product-space contrastive framework (ProCLIP) on image–text retrieval tasks. Our experiments are designed to isolate the effect of representation geometry on multimodal alignment, while controlling for backbone architecture, training data, and optimization settings. In particular, we ask whether distributing representational capacity across complementary geometries yields stronger and more stable alignment than committing to a single manifold.

### 3.1 DATASETS AND EVALUATION TASKS

We use two standard benchmarks for image–text alignment and retrieval: Flickr30k and MSCOCO. Flickr30k contains 31K images, while MSCOCO provides approximately 123K images, and both datasets include five captions annotated for each image. We conduct experiments and report bidirectional retrieval performance (text-to-image and image-to-text) using standard Accuracy@1, 5, and 10 metrics. For image-to-text tasks, we consider a retrieval successful if at least one of the five captions appears within the specified top-$k$ bucket.

### 3.2 FEATURE EXTRACTION AND EXPERIMENTAL PROTOCOL

All experiments are conducted using frozen CLIP (Radford et al., 2021) representations to isolate the effect of geometric alignment from encoder finetuning. We use the ViT-B/32 CLIP model to extract pooled image and text embeddings. For each dataset, embeddings are L2-normalized and cached prior to training, ensuring that all models operate on identical input features.

ProCLIP extends these frozen embeddings using lightweight geometry-specific projection heads that map image and text representations into a mixed-curvature product embedding space. As illustrated in Figure 2, image and text inputs are first encoded by frozen CLIP visual and text encoders to produce modality-specific features. These features are then passed through three parallel projection heads corresponding to hyperbolic, spherical, and Euclidean components. Each head maps the CLIP feature into its respective geometric space, producing embeddings $(z^{(H)}, z^{(S)}, z^{(E)})$. The resulting components are combined to form a product-space representation, and cross-modal similarity is computed using the product-space distance defined in Section 2. Each component has dimensionality $d$, resulting in a total nominal dimensionality of $3d$ for the product-space model.

For comparison, we also train three single-geometry baselines—Euclidean-only, spherical-only, and hyperbolic-only—using identical projection-head architectures and optimization settings, each with dimensionality $3d$ to ensure fair comparisons.

### 3.3 TRAINING OBJECTIVE AND OPTIMIZATION

All models are trained using a symmetric contrastive loss, as described in previous sections, which jointly enforces image-to-text and text-to-image alignment within each minibatch. During training, we randomly select one caption per image in each epoch. Optimization is performed in feature space using AdamW with an optional cosine learning-rate scheduler. No data augmentation or backbone finetuning is employed. Model selection is based on validation Accuracy@1, averaged across image-to-text and text-to-image tasks, and final evaluation is performed on held-out test splits.

**Training details.** Projection heads are implemented as two-layer MLPs with hidden dimension 512 and ReLU activation. Models are trained using AdamW with learning rate $1 \times 10^{-4}$ and weight decay $1 \times 10^{-4}$. The batch size is set to 1024, and the temperature parameter $\tau$ in the contrastive loss is 0.07. We apply a cosine learning-rate schedule with linear warmup for the first 5 epochs. All experiments are conducted on a single NVIDIA RTX A6000 GPU with 48GB memory.

### 3.4 RESULTS

We report image-to-text and text-to-image retrieval results in Table 1. ProCLIP consistently outperforms single-manifold models across all embedding dimensionalities and in both retrieval directions. The use of mixed curvature improves retrieval Accuracy@1 by 2–7 percentage points, depending on the dataset and embedding dimensionality.

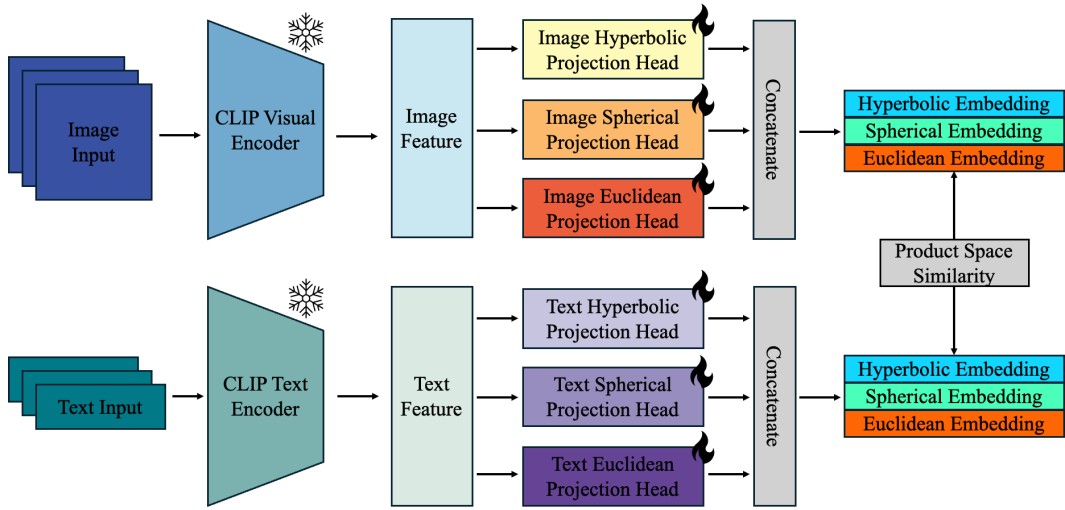

Figure 2: **Architecture of ProCLIP**. Image and text inputs are first encoded using frozen CLIP visual and text encoders to produce modality-specific features. Lightweight geometry-specific projection heads then map these features into three complementary geometric components: hyperbolic, spherical, and Euclidean embeddings. The resulting representations are combined into a mixed-curvature product space, and cross-modal alignment is computed using a product-space similarity within a CLIP-style contrastive learning framework.

Table 1: **Retrieval on Flickr30k and MSCOCO** (Recall@K). We report R@1 and R@10 for image-to-text and text-to-image. Best results within each dataset and dimension block are in **bold**.

| Dim. | Model | Flickr30k | | | | MSCOCO | | | |
|---|---|---|---|---|---|---|---|---|---|
| | | Image → Text | | Text → Image | | Image → Text | | Text → Image | |
| | | R@1 | R@10 | R@1 | R@10 | R@1 | R@10 | R@1 | R@10 |
| 64 × 3 | ProCLIP | **0.8080** | 0.9780 | **0.6428** | **0.9380** | **0.6198** | **0.9266** | **0.4703** | **0.8344** |
| | Spherical | 0.7740 | **0.9810** | 0.6392 | 0.9354 | 0.5962 | 0.9136 | 0.4448 | 0.8284 |
| | Hyperbolic | 0.7390 | 0.9720 | 0.6024 | 0.9164 | 0.5952 | 0.9094 | 0.4388 | 0.8064 |
| | Euclidean | 0.6890 | 0.9460 | 0.5432 | 0.8914 | 0.5326 | 0.8756 | 0.3886 | 0.7718 |
| 128 × 3 | ProCLIP | **0.8070** | **0.9880** | **0.6634** | **0.9386** | **0.6448** | **0.9282** | **0.4849** | **0.8446** |
| | Spherical | 0.7770 | 0.9830 | 0.6480 | 0.9360 | 0.5856 | 0.9152 | 0.4439 | 0.8258 |
| | Hyperbolic | 0.7860 | 0.9760 | 0.6298 | 0.9270 | 0.5932 | 0.9070 | 0.4387 | 0.8086 |
| | Euclidean | 0.7220 | 0.9550 | 0.5950 | 0.9064 | 0.5548 | 0.8818 | 0.4013 | 0.7830 |
| 256 × 3 | ProCLIP | **0.8320** | **0.9870** | **0.6752** | **0.9496** | **0.6696** | **0.9408** | **0.5072** | **0.8578** |
| | Spherical | 0.7710 | 0.9820 | 0.6498 | 0.9344 | 0.5970 | 0.9186 | 0.4479 | 0.8296 |
| | Hyperbolic | 0.7780 | 0.9780 | 0.6400 | 0.9324 | 0.5990 | 0.9096 | 0.4446 | 0.7172 |
| | Euclidean | 0.7480 | 0.9600 | 0.5994 | 0.9134 | 0.5650 | 0.8900 | 0.4114 | 0.7905 |

## 4 CONCLUSION AND FUTURE WORK

We proposed a geometry-aware mixed-curvature product space for contrastive multimodal alignment and demonstrated consistent improvements over single-manifold baselines on image–text retrieval benchmarks. The product construction offers a flexible representation space that can accommodate heterogeneous semantic structure through complementary geometric components. Promising directions for future work include learning more fine-grained multimodal representations that capture both intra-modal structure and cross-modal correspondences, as well as developing theoretical analyses that characterize when and why mixed-curvature product spaces yield advantages over single-geometry embeddings.

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
