# OpenReview forum: "ProCLIP: Product Space Multimodal Contrastive Alignment"
_ICLR.cc/2026/Workshop/GRaM — ICLR 2026 Workshop GRaM Poster_

### Official Review · Reviewer_PaaN · 2026-02-24
**Review of ProCLIP**

**Rating:** 5
**Confidence:** 4

**Review:**

The article argues that standard contrastive learning models like CLIP use a single embedding geometry (e.g., Euclidean/spherical), which struggles with mixed semantic structures in multimodal data (e.g., hierarchies in image-text pairs). It proposes ProCLIP (short for Product space CLIP), a drop-in extension that embeds data into a mixed-curvature product space combining hyperbolic (for hierarchies), Euclidean (for linear variations), and spherical (for directions) manifolds. Similarity is measured using a weighted product metric, and training uses a CLIP-style contrastive loss with lightweight projection heads applied to the frozen CLIP features. Evaluated on image-text retrieval benchmarks (Flickr30k, MSCOCO), it shows 2-7% gains in recall over single-manifold baselines.

**Positives:**

- *Clever Geometric Innovation:*  Effectively combines curvatures to handle diverse semantics (e.g., hierarchical like "animal > dog"), providing a flexible inductive bias without major architectural changes.

- *Better Empirical Results:* Consistent improvements in retrieval metrics (e.g., R@1 up to 0.832 vs. baselines ~0.75-0.78 on Flickr30k), with ablations showing mixed space outperforms single ones across dimensions.

- *Practical and Efficient:* Drop-in replacement for cosine similarity; uses frozen backbones, minimal compute overhead, and learnable weights that adapt curvatures implicitly.

- *Interpretive Insights:* Analyses highlight where hyperbolic components shine (e.g., nested concepts), aligning well with the workshop's geometry focus.

- *Principled Extension:* Builds on prior work (e.g., `Gu et al. 2018`) but applies to multimodal, offering a tidy, motivated upgrade for VLMs.

**Negatives:**

- *Limited Novelty:* Feels incremental—heavily inspired by HyCoCLIP (`Pal et al. 2024`) and single-modality product spaces; the core idea is a straightforward combo rather than a breakthrough.

- *Narrow Scope:* Only tested on image-text retrieval with basic benchmarks; no video-audio, zero-shot transfer, robustness to noise, or real-world datasets.

- *Not enough details:*  Lacks deep ablations (e.g., on weight learning, curvature sensitivity) or theoretical analysis; implementation specifics (e.g., optimization hyperparameters) are sparse.

- *Potential Overfitting to Hierarchies:* Gains emphasized in hierarchical data, but unclear performance in non-hierarchical regimes.

- *No Broader Comparisons:*  Misses evals against recent rivals like Lorentzian CLIP (L-CLIP) or entailment-focused models.

**Pmlr Suitability:**

NA

---

### Official Review · Reviewer_32T2 · 2026-02-24
**Review of ProCLIP paper**

**Rating:** 5
**Confidence:** 5

**Review:**

This paper proposes a mixed-curvature product manifold for contrastive multimodal learning, extending frameworks such as CLIP beyond the standard single-geometry assumption. The idea is conceptually well-motivated and empirically shows consistent improvements on image–text retrieval benchmarks. The geometry-aware objective is clean and integrates naturally into existing pipelines. However, the work lacks sufficient implementation details (e.g., curvature parameterization, optimization stability, computational overhead), which may limit reproducibility. In addition, the theoretical justification remains largely intuitive, with limited formal analysis of why mixed curvature improves generalization or alignment. Stronger technical transparency and deeper theoretical grounding would significantly strengthen the contribution.

## Pros
1. **Strong Motivation**: Identifies limitations of single-manifold contrastive learning
2. **Conceptual Clarity**: Intuitive explanation of heterogeneous multimodal semantics
3. **Empirical Gains**: Consistent improvements on image–text retrieval benchmarks
4. **Practical Design**: Drop-in compatibility with CLIP-style frameworks

## Cons
1. **Implementation Details Lacking**: Curvature parameterization, optimization stability, and computational cost are under-specified
2. **Limited Theoretical Analysis**: Only intuitive justification; no formal explanation of why mixed curvature improves alignment
3. **Incremental Novelty**: Builds on existing geometric embeddings, main contribution is integration
4. **Task Scope Limited**: Focuses on retrieval, unclear effect on other multimodal tasks

**Pmlr Suitability:**

NA

---

### Official Review · Reviewer_S3WN · 2026-02-24
**Review of ProCLIP: Product Space Multimodal Contrastive Alignment**

**Rating:** 6
**Confidence:** 3

**Review:**

This paper argues that multimodal semantics are heterogeneous, incapable of being modeled well within one unified euclidean space, which is what might be currently done with a CLIP-like model. Instead, this paper proposes to use a latent space that can support multiple types of structure by combining different geometries, building on the idea that mixed-curvature product spaces can model heterogeneous relations better. They define a product manifold over Hyperbolic Euclidean, and Spherical spaces, and there is a weighted product across each factor to produce the final embedding distance. They begin from a CLIP model and add projection heads, acting as a drop-in replacement for cosine similarity. The authors demonstrate improved results for image-text retrieval compared to the Euclidean-only or other combination models. Overall the paper's idea is simple and demonstrated within a limited experimental setting, but could provide further inspiration for others interested in the representation surface of multimodal embedding models.

Strengths
- The proposed method is a relatively simple addition, and gives a way to allocate geometry to different semantic structures, importantly fitting on top of existing pretrained models.
- The combination weights and formulation are motivated well enough (in terms of why they could possibly be useful).

Weaknesses
- The paper lacks any analysis connecting the original hypothesis of the heterogenous space being better for representing multimodal data with the experimental results. The results are better, but it is not clear why. Do the results line up with the hypothesis? It's unclear as of now.
- The implications are quite limited when the only training that is done is on the last head, not the model itself end to end.
- How do these heads help with other downstream tasks of CLIP?
- Seeing other combinations of the structures as additional ablations would be interesting.
- The paper has no figure unfortunately, it could be quite helpful to build intuition or show off results.

Overall it would be great to see further analysis and experiments, such that the claims can be actually substantiated with the evidence in a more specific way, not necessarily just that the performance increases a few points.

**Pmlr Suitability:**

NA

---

### Meta-Review · Area_Chair_MnEd · 2026-02-25

**Decision:**

Accept

**Metareview:**

This paper's topic is in-scope for the workshop, and the idea is original and likely to be of interest for the GRaM community. The method is simple but shows preliminary empirical gains, although extending to other tasks and including other baselines would strengthen the experiments section. The theoretical analysis is lacking, according to reviewers. The authors should consider and address the reviewers' criticisms, but for a tiny paper (representing an in-progress idea), I recommend acceptance.

**Relevance To Proceedings:**

Tiny paper — does not apply

**Relevance To Workshop:**

Yes — suitable for GRaM

---

### Decision · Program_Chairs · 2026-03-02

Accept (Poster)